# Landscape structure shapes the diversity of tree seedlings at multiple spatial scales in a fragmented tropical rainforest

Sergio Nicasio-Arzeta[1,2☉]*, Isela E. Zermeño-Hernández[3‡], Susana Maza-Villalobos[4‡], Julieta Benítez-Malvido[2☉]

**1** Programa de Doctorado en Ciencias Biomédicas, Universidad Nacional Autónoma de México (UNAM), Ciudad de México, Morelia, México, **2** Instituto de Investigaciones en Ecosistemas y Sustentabilidad, Universidad Nacional Autónoma de México, Morelia, Michoacán, México, **3** CONACyT-Universidad Michoacana de San Nicolás de Hidalgo, Morelia, Michoacán, Mexico, **4** CONACyT-El Colegio de la Frontera Sur, Tapachula, Chiapas, México

☉ These authors contributed equally to this work.
‡ These authors also contributed equally to this work.
* sergio.nicasio@iies.unam.mx

**Data Availability Statement:** The data of seedling abundance and landscape metrics have been deposited with the Dryad Data Repository

## Abstract

The maintenance of seedling diversity of animal-dispersed tree species is fundamental for the structure and function of forest patches in fragmented tropical rainforests. Nonetheless, the effects of landscape structure at different spatial scales on α- and β-diversity of tree seedling communities are recently explored. Using a multi-scale approach, we assessed the relative effect of landscape composition and configuration on α- and β-diversity of animal-dispersed seedlings within 16 forest patches in the Lacandona rainforest, Mexico. We assessed these effects at 13 spatial scales (from 300 to 1500 m radius, at 100 m intervals) for three metrics of effective number of species considering α- and β-diversity. We found that α-diversity was largely affected by landscape composition and β-diversity by landscape configuration. On the one hand, the amount of secondary forest influenced α-diversity. Additionally, species richness increased in landscapes with highly aggregated forest patches. On the other hand, β-diversity was affected positively by forest fragmentation and negatively by the edge contrast of forest patches with the surrounding matrix. Our findings indicate that landscape configuration is a strong driver of seedling diversity in highly deforested rainforests. Promoting forest patches and secondary forests through payment for ecosystem services' programs, favoring matrix quality within land-sharing schemes of smallholder agriculture and secondary forest management, and identifying restoration opportunities for assisted or unassisted natural regeneration are urgently needed for conservation of seedling diversity in human-modified tropical landscapes.

## Introduction

Tropical rainforests harbor half of known animal and plant species of the planet [1], yet accelerated forest conversion to cattle pastures and large-scale plantations is restraining them to

(https://www.datadryad.org/, https://doi.org/10.5061/dryad.g1jwstqr1).

**Funding:** J.B.M. received grant projects from Programa de Apoyo a Proyectos de Investigación e Innovación Tecnológica (PAPIIT), Dirección General de Asuntos del Personal Académico UNAM (IN214014, IN202117 and IN201620), and from Consejo Nacional de Ciencia y Tecnología (CONACyT), México (CB2005-C01-51043, CB2006-56799 and CB2007-7912). S.N.A. is a doctoral student from Programa de Doctorado en Ciencias Biomédicas, Universidad Nacional Autónoma de México (UNAM) and received CONACyT fellowship 317569. The funders had no role in study design, data collection and analysis, decision to publish, or preparation of the manuscript.

**Competing interests:** The authors have declared that no competing interests exist.

forests patches embedded in a matrix of anthropogenic land covers (known as human-modified tropical landscapes, or HMTLs [2,3]). Forest patches of HMTLs are generally small (mean size = 13–17 ha), and their number are expected to rise in the following years [4].Yet, these patches can preserve a large number of plants and animals species [5,6], and contribute to improve forest regeneration, landscape connectivity, metapopulation persistence, and resource availability for native species [7,8]. These contributions are strongly influenced by patches' tree community, which provides of food and habitat for an important proportion of the old-growth fauna, and drive biomass production and carbon storage [9,10]. Thus, sustaining forest regeneration within patches is critical to promote the long-term maintenance of biodiversity in HMTLs [11,12].

The future composition and structure of tropical rainforests is driven by the seedling community, which makes its assessment helpful to comprehend the forthcoming regeneration of forest patches [13–15]. Seedling communities are characterized for a high local species richness (α-diversity) and an impressive species turnover (β-diversity), which are shaped by seed dispersal, environmental heterogeneity and density dependence effects [16–19]. Disruption of these processes limit seed dispersal/germination and seedling establishment of the animal-dispersed species (which represents nearly 90% of tree species [20]), leading to the richness decline and biotic homogenization of seedling communities [21–23]. As result, the regenerating forests are dominated by generalist and abiotically-dispersed tree species, supports a lower density of saplings, show reduced recruitment of animal-dispersed species, and their composition and structure resemble to secondary forests [24–26]. Although a growing body of conservation strategies have emerged in order to preserve plant diversity in the tropics [12,27,28], none of them refer to seedling communities in HMTLs, since the effects of the types and amount of land covers in the landscape (landscape composition), and their spatial arrangement (landscape configuration) on the α- and β-diversity are poorly understood (but see [29,30]).

Evidence suggests that seedling α-diversity is affected by landscape composition, while β-diversity is more influenced by landscape configuration. Recent studies have shown that reduction of α-diversity in seed and sapling assemblages by forest loss and dominance of open matrices (landscape composition) is associated with limitations in local seed sources, seed dispersal and sapling establishment of animal-dispersed species [10,30,31]. Additionally, seedling α-diversity may also be influenced by seed sources from nearby secondary forests [31–33]. Finally, the effects of landscape configuration on α-diversity were negligible [30]. On the contrary, seedling β-diversity within patches may be linked to forest fragmentation and edge effects (landscape configuration). Firstly, β-diversity is promoted when habitat is spread out in several patches than within a single large patch, whether by random sampling (similar patches sample different species from a regional pool) or by greater environmental variations among disaggregated patches [34,35]. Secondly, edge effects are known to limit the establishment of animal-dispersed seedlings commonly found in old-growth forests [13,36], contributing to β-diversity loss of tree assemblages [25]. Moreover, edge contrast can indirectly affect β-diversity by influencing the presence of terrestrial mammals within patches [6,37], which shape the composition of seedling communities [22–24]. These effects are expected to be stronger in highly deforested regions, particularly with <30% of forest cover (fragmentation threshold hypothesis) [38].

Assessing landscape effects on tree seedling diversity is challenging, because these effects vary on the spatial scale at which landscape composition and configuration are analyzed [39,40]. The extent of the area relevant on particular biological responses (i.e. the scale of effect [41]) varies greatly among the landscape compositional and configurational patterns measured [42,43]. In addition, the scale of effect also differs by the type of biological response assessed,

since each response is influenced at different spatial and temporal scales [39,42]. For example, it is hypothesized that biological variables shaped by species colonization and extinction dynamics (i.e. species occurrence) should have larger scales of effect than those affected by local drivers (i.e. species abundance) [43]. Thus, the scale of effect is expected to be larger for lower-density populations (rare species) than for higher-density populations (typical or dominant species) [42]. This is particularly important in tropical rainforests, where a high number of species coexist at very low-density populations, making seedling communities highly α- and β-diverse in space [17,44].

Multi-scale approaches are a hierarchical approach useful to increase the possibility to detect the scale of effect of landscape structure on biological responses [45,46]. In these approaches, it is necessary the use of a wide range of scales, as assessments performed using very few scales within narrow ranges overlook the scale of effect [41,45]. To our knowledge, studies that have employed multi-scale analysis on plant communities in the tropics are scant [30,47], and none of them have assessed the effect of landscape structure on seedling communities in HMTLs (but see [30]). In this study, we employed a multi-scale approach to assess the contribution of landscape composition and configuration on tree seedling α- and β-diversity within rainforest patches. We particularly focused on the following question: Which components of landscape structure are influencing seedling α- and β-diversity and at which spatial scales?

Furthermore, we tested the following predictions: (1) α-diversity will increase in landscapes with high forest cover and β-diversity will increase in fragmented landscapes with low edge contrast; and (2) there should be larger scales of effect for richness-based metrics than for abundance-based metrics of seedling α- and β-diversity (i.e. dominant < typical < all species).

## Material and methods

### Study site

We conducted the study at the Marqués de Comillas region in the Lacandona rainforest in southeastern Mexico (Fig 1A). The monthly temperature oscillates between 24 and 26˚C, and the annual precipitation ranges from 2500 to 3500 mm [48]. This region encompasses the largest rainforest of the Mesoamerican biodiversity hotspot and the Montes Azules and Lacantún Biosphere Reserves, which protect 393,000 ha of old-growth forest [49–51]. Land-cover change to cattle pastures has reduced its forest cover outside the reserve by more than 50% of its original extension [51,52]. Despite deforestation in the last 40 years, large mammals, birds, and primate species are still present in forest patches [6,53,54]. We selected 16 old-growth forest patches, ranging from 1 ha to 63 ha (Fig 1B). We only considered only patches located in floodplains to keep adult tree species composition, soil type and abiotic factors relatively constant. The land tenure of the Marqués de Comillas region is private, and land property is distributed among the inhabitants [51,52]. Therefore, we conducted our study only in those forest patches where permission was granted by the owners.

### Tree seedling sampling

Patches were sampled by 1-ha blocks placed at the center of each forest patch from February to June of 2014 (Fig 1C). Each block contained ten 1-$m^2$ plots randomly arranged in groups of two or three plots along five equidistant transects (20 m apart; Fig 1D). For 1-ha forest patches, we never positioned plots at the edge in order to have a 20-m buffer zone of vegetation that protect them from edge effects as much as possible (Fig 1D) [13,15]. Within each 1-$m^2$ plot we counted and identified all tree seedlings (10–100 cm height) to the lowest possible taxonomic level with the help of a local parataxonomist and field guides [56,57]. When field identification

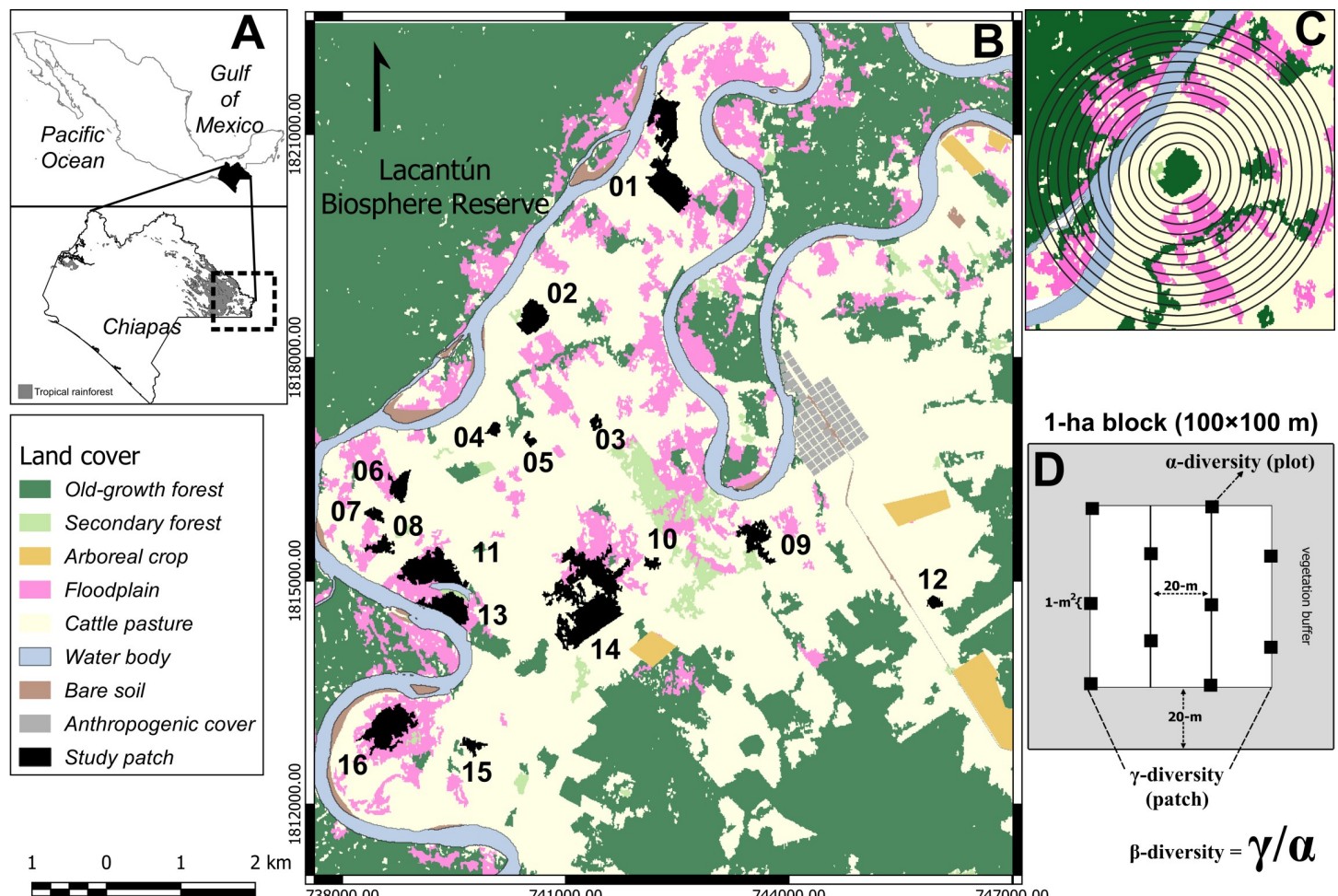

**Fig 1. Study site and sampling design of tree seedlings.** The figure shows the location of the study area (A) and the 16 study patches (black) in the Marqués de Comillas region (B). The 13 buffer sizes around the geographic center of the sampling block (C). Sampling method of tree seedlings (see material and methods), and the source of γ-diversity (species located in the 1-ha block), α-diversity (the average number of species per 1-m2 plot) and β-diversity (the effective number of completely distinct assemblages within each 1-ha block) (D). Reprinted from [55] under a CC BY license, with permission from Sergio Nicasio-Arzeta, original copyright 2019.

was not possible, we took samples for identification at several herbariums (MEXU, ECO-SC-H). We determined the dispersal syndrome of each species, based on their fruits and seed morphology [58,59]. This study did not involve the extraction or damage of endangered species. We based our analyses on animal-dispersed tree species only because they comprise up to 90% of the seed rain in rainforests [20,60]. In our case, 94% of the seedling species were animal-dispersed (see Results). Plant nomenclature followed the Missouri Botanical Garden database Tropicos [61].

### Diversity estimation

We assessed the sampling completeness with the sample coverage estimator of Chao and Shen [62]. We combined the data of the 10 sampling plots within each patch. Then, we estimated the proportion of the total number of individuals that belong to the species represented in the sample [63]. The sample coverage among patches was high (91.07 ± 7.47%; mean ± SD), indicating that our sampling effort was adequate for estimating seedling species diversity [64]. We then estimated the effective number of species for seedlings using the diversity decomposition

of effective numbers of species (the Hill numbers [65,66]). This method assesses the relative importance of species occurrence and abundance on α- and β-diversity [65,67]. Assessing these diversity metrics is important in the tropics because most tree species have small populations, and the scale of effect of landscape structure is expected to differ among species with varying abundances [60,68]. Consequently, diversity metrics that account for species abundance are expected to have smaller scales of effect than those measured by species occurrence only (presence-absence) [42,43]. Hill numbers ($^q D$) are in units of species, which permit the characterization of species abundance distribution of a community and provide complete information about the community diversity [66,67]. We first calculated the patch gamma (γ) diversity of order $q$ as follows:

$$^q D_\gamma = (\sum_{i=1}^{S} \bar{p}_i^q)^{\frac{1}{(1-q)}}$$

where $\bar{p}_i$ denotes the mean relative abundance of the $i$th species in the $N$ 1-m$^2$ plots [65,69], and $q$ is a parameter that determines the sensitivity of the measure to the relative abundances. Since this measure is undefined for q = 1, the γ-diversity of order one can be estimated with the following formula:

$$^1 D_\gamma = exp(-\sum_{i=1}^{S} \bar{p}_i \log \bar{p}_i)$$

The $^0 D$ represents the species richness (hereafter called all species), which is not sensitive to individual abundances [65,69]. The $^1 D$ (equivalent to the exponential of Shannon's entropy index) weights each species according to its abundance in the community, and could be interpreted as the number of equally common species which does not favor rare nor abundant species (hereafter called typical species) [65]. Finally, the $^2 D$ (equivalent to the inverse Simpson index) gives a high weight to abundant species and could be understood as the effective number of the most abundant species (hereafter called dominant species) [67,70]. We then considered the 1-m$^2$ plots to calculate the alpha (α) diversity with the formula:

$$^q D_\alpha = \left( \frac{1}{N} \sum_{i=1}^{S} p_{i1}^q + \frac{1}{N} \sum_{i=1}^{S} p_{i2}^q + \cdots \right)^{\frac{1}{(1-q)}}$$

Where $p_i$ denotes the relative abundance of the $i$th species in each of the $N$ 1-m$^2$ plots. We estimated α-diversity when q = 1 as:

$$^1 D_\alpha = exp \left\{ -\frac{1}{N} \left( \sum_{i=1}^{S} (p_{i1}^q \ln p_{i1}^q) + \sum_{i=1}^{S} (p_{i2}^q \ln p_{i2}^q) + \cdots \right) \right\}$$

Afterward, we used the resulting α$_{plot}$- and γ$_{patch}$-diversity to calculate the "effective number of completely distinct assemblages" within each patch (β-diversity) as follows [65]:

$$^q \beta_{plot} = \frac{^q \gamma_{patch}}{^q \alpha_{plot}}$$

This β-diversity ranges between one (when the assemblages of all 1-m$^2$ plots are identical) and $N$ (when the assemblages of the $N$ 1-m$^2$ plots are completely different from each other). We used the package *vegan* in R [71].

## Landscape metrics and multi-scale assessment

We employed a multispectral SPOT-5 satellite image of 10 × 10 m pixel resolution recorded in March 2013 to carry out a supervised classification using the GRASS GIS software [72]. We

**Table 1. Description, metric type and ecological relevance of the landscape metrics measured at class level employed in the study.**

| Metric | Description | Metric type | Ecological relevance | References |
|---|---|---|---|---|
| Forest cover (FC) | Percentage of landscape area covered by old-growth forest | Composition | Indicator of the landscape-scale habitat amount. It is positively associated with availability of propagules and seed dispersal | [31,60] |
| Secondary forest (SF) | Percentage of landscape area covered by secondary forest (regrowth of ≤15 years) | Composition | Measurement related to impacts on the dynamics and trajectories of floristic change within forest patches | [32,74] |
| Patch density (PD) | Number of forest patches per landscape area (n/ha) | Configuration | Fragmentation metric associated with increasing edge effects, the number of seed sources in the landscape and with landscape connectivity of seed dispersers in the tropics | [47,75] |
| Aggregation index (AI) | Percentage of like-adjacencies between forest patches. Maximum aggregation indicates a single, compact patch | Configuration | Aggregated patches facilitate the inter-patch movement, promoting dispersal at smaller spatial scales. These conditions allow the persistence of species in highly fragmented landscapes through complementation/supplementation dynamics | [8,76,77] |
| Patch isolation (PI) | Mean distance between forest patches within the landscape | Configuration | Metric employed to estimate connectivity on seed dispersal, altering seed abundance/richness and floristic differentiation among patches | [31,78] |
| Edge contrast index (EC) | Average degree of edge contrast between forest patches and their immediate neighborhoods | Configuration | High percentages of contrasting edges reduce the connectivity of vertebrate seed dispersers and terrestrial mammals in tropical forests | [6,79] |

used sampling points representing seven land-cover classes, including the following: 1) old-growth forest (undisturbed rainforest), 2) secondary forest (rainforest regrowth), 3) flood-plains (swamp forests of bamboo and palm trees subjected to periodic flooding), 4) arboreal crops (oil-palm plantations), 5) cattle pasture (induced grassland for cattle raising), 6) anthropogenic cover (roads and urban settlements), and 7) water bodies (rivers and permanently flooded areas). The overall classification accuracy was 79%.

We then calculated six landscape metrics (Table 1) considered as drivers of seed and seedling community diversity [47,60]. The composition metrics included the percentages of old-growth forest (FC) and secondary forest (SF) covers, whereas the configuration metrics were the number of forest patches per landscape area (patch density; PD), the adjacency among forest patches (aggregation index; AI), the mean distance among forest patches (patch isolation; PI), and the edge contrast index (EC). We calculated the EC using quality values for matrix covers that describe both their capacity to reduce edge effects and the permeability for terrestrial mammals' movement. These values were based on the assumption that edge effects increase and mammals' presence declines along a gradient of habitat loss and relates the percentage of each land-cover type within the landscape matrix to its relative quality [2,6]. We ranked the relative quality of each land-cover type based on the suitability of vegetation structure for regulating edge microclimate and for mammal feeding, movement and/or habitat on a seven-point scale, including the following: 1 (water bodies, with the lowest suitability); 2 (anthropogenic cover); 3 (cattle pasture); 4 (arboreal crops); 5 (floodplains); 6 (secondary forest); and 7 (old-growth forest, representing the highest suitability). To obtain a more robust and realistic representation of landscape structure effects, we estimated the area-weighted mean of the EC index [73].

We neither considered patch size nor patch shape because both variables were highly correlated ($r = 0.81$), and none of them had significant effects on any of the α- and β-diversity metrics (Table A in S1 Text).

We followed the recommendation about the landscape radius that best predicts biological responses, which is 0.3–0.5 times the maximum dispersal distance of seed dispersers and herbivore mammals foraging behavior [41,45]. The maximum dispersal distance of arboreal seed dispersers and terrestrial mammals in the study region ranges between 500 and 4000 m [54,80–82].Thus, we estimated six landscape metrics within 13 circular buffers (300 to 1500 m

radius, at 100 m intervals) from the center of each focal patch (Fig 1C). We only found correlations between AI and FC between the 900–1500 m radius (Table C in S1 Text). The distance among sampling sites was of 3602.56 ± 1932.21 m (mean ± standard deviation), and were randomly aggregated (Clark-Evans $R$ = 1.02; $p$ = 0.86) [83]. The overlapping between buffers of nearby sampling points increased with buffer sizes (Table D in S1 Text). However, we did not find spatial autocorrelation between the distance of sampling sites and the diversity metrics (Table E in S1 Text), nor between the distance of sampling sites and the landscape metrics across the 13 buffer sizes (Table F in S1 Text).

### Statistical analyses

Firstly, we estimated the scale of effect of each landscape metric using linear models. We fitted a diversity metric with a single landscape metric. We repeated this process on each buffer size and obtained 13 linear models of the same landscape metric. We verified variable normality with a Shapiro-Wilk test [84]. Then, we assessed the predictive power of each model using a leave-two-out cross-validation. This analysis splits the dataset in two sections; the first split contains the diversity data from 14 patches as calibration data and fits a linear model which is employed to predict the diversity values of the second split, which contains the two remaining values (validation data). We calculated the sums of squares between the estimated and observed diversity values employed as validation data. We repeated this procedure for each possible split of the data set and calculated the average sum of squares ($\bar{SS}_{cv}$). Next, we employed the $\bar{SS}_{cv}$ to calculate the proportion of the variation that can be predicted by the model using the leave-two-out coefficient of determination ($R^2_{CV}$) as follows:

$$R^2_{CV} = 1 - \frac{\bar{SS}_{cv}}{\frac{1}{n}\sum_{i=1}^{n}(y_i - \bar{y})^2}$$

where $\bar{SS}_{cv}$ is the average sum of squares obtained from the leave-two-out cross-validation, $y_i$ is the diversity value for the $i^{th}$ patch, $\bar{y}$ is the mean value of a given diversity metric, and $n$ is the number of patches. The $R^2_{CV}$ ranges between $-\infty$ (indicating the model has a worse prediction power than the null model) and one (indicating the model predicts the validation data perfectly), and can be used to compare between response variables and scales of measurement [85–87]. Finally, we plotted the resulting 13 $R^2_{CV}$ values and selected the one with the strongest response as the scale of effect for that landscape metric (Figs A and B in S1 Text [45]). We repeated this procedure for each diversity and landscape metric ($N$ = 468 models).

Flawed study design, such as selecting a narrow range of spatial extents (smallest to largest) or by establishing wide distances among the analyzed extents, can lead to inaccurate estimations of the scale of effect [41,43]. Therefore, we estimated the uncertainty around the selected scale of effect through bootstrapping [88]. We randomly resampled the data from $n$ patches, with resampling, from the set of 16 patches 1000 times. Then, we estimated the scale of effect for each resampled data set as described above and summed the number of times (out of 1000) that each buffer size was selected as the scale of effect. Although the uncertainty around the selected scale of effect varied among landscape- and diversity-metrics, the most frequently selected scale of effect based on the bootstrapped analysis matched with the scale of effect that we selected based on the highest $R^2_{CV}$ value (Figs C and D in S1 Text). Additionally, we employed the data from the bootstrapped analysis to assess if the scale of effect differed among α- and β-diversity metrics and among landscape metrics. We only employed the data from the selected landscape metrics in the previous step.

Thereafter, we evaluated the effects of landscape metrics on diversity metrics through multiple linear models with a normal distribution. We constructed a global model for each of the

**Table 2. Global model of each diversity metric employed for the model selection assessment.**

| α-diversity | β-diversity |
|---|---|
| $\alpha_{all} \sim AI_{600} + PI_{1200} + SF_{600}$ | $\beta_{all} \sim EC_{800} + PD_{1400}$ |
| $\alpha_{typical} \sim AI_{600} + PI_{1200} + SF_{600}$ | $\beta_{typical} \sim PD_{500} + SF_{1300}$ |
| $\alpha_{dominant} \sim AI_{600} + PI_{1200} + SF_{600}$ | $\beta_{dominant} \sim AI_{500} + PD_{500}$ |

The landscape metrics are the aggregation index (AI), the edge contrast index (EC), patch isolation (PI), and the percentage of secondary forest (SF). The subscript numbers indicate the scale of effect (radius meters) of each landscape metric.

α- and β-diversity metrics using only the landscape metrics at the scale of effect identified in the previous step (Table 2). All models were additive due to our limited sample size of 16 forest patches.

We did not find spatial autocorrelation between the distance of sampling sites and the model residuals of the selected landscape metrics (Table G in S1 Text). Therefore, we assured samples independency because spatial autocorrelation of model residuals–an indicator of pseudoreplication–is associated with the proximity between sampling sites [89,90]. We estimated the variance inflation factor (VIF) of landscape metrics beforehand with the *car* package [91]. Since we did not find significant collinearity (VIF ≥ 4) between the explanatory variables [92], we employed the *dredge* function of the *MuMIn* package [93] to create all possible combinations of explanatory variables plus the null model (only the intercept). We ranked the models using the Akaike's information criterion corrected for small samples (AICc) and selected those models with a AICc difference lower than two (ΔAICc < 2) as the best supported by the data [94].

Finally, we assessed the importance and the relative effect of each landscape metric measured at the scale of effect on each diversity value using an information theory approach and multimodel inference [94]. For this, we selected the subset of models that had 95% probability of containing the best model using the summed Akaike weights ($w_i$) of ranked models until $\Sigma w_i \leq 0.95$. We employed the $w_i$ of the model's subset to calculate the relative importance and the model-averaged parameter estimates of each explanatory variable. We considered as influential variable those for which the unconditional standard error (USE) did not include zero in the averaged parameters. We carried out all statistical analysis in the R 3.5.2 statistical computing environment [95].

## Results

We recorded a total of 1334 tree seedlings from 29 families, 51 genera and 72 species in 160 m$^2$ (Table H in S1 Text). Most seedlings were animal-dispersed species (1258 individuals; 94.3%), belonging to 24 families, 42 genera and 58 species. Mean species density of the animal-dispersed seedlings were 11.37 ± 2.96 species/10 m$^2$ (range 6–16 species; mean ± standard error), mean seedling density was 78.62 ± 34.03 individuals/10 m$^2$ (range 10–129 individuals). The most abundant species was *Inga punctata*,—an early successional and disturbance tolerant tree species -, which represented 33% of all individuals sampled, followed by the late-successional tree species *Ampelocera hottlei* (13%) and *Brosimum alicastrum* (11%). Less than half of the species (ca. 31%) were restricted to one patch. We found that α- and β-diversity metrics were significantly higher than the other metrics when species abundances were not considered (Fig E in S1 Text).

### Scale of effect and importance of landscape structure on seedling diversity

We observed that scale of effect of landscape composition and configuration was similar for all α-diversity metrics (Fig 2A–2C). The bootstrapped data showed the scale of effect of α-

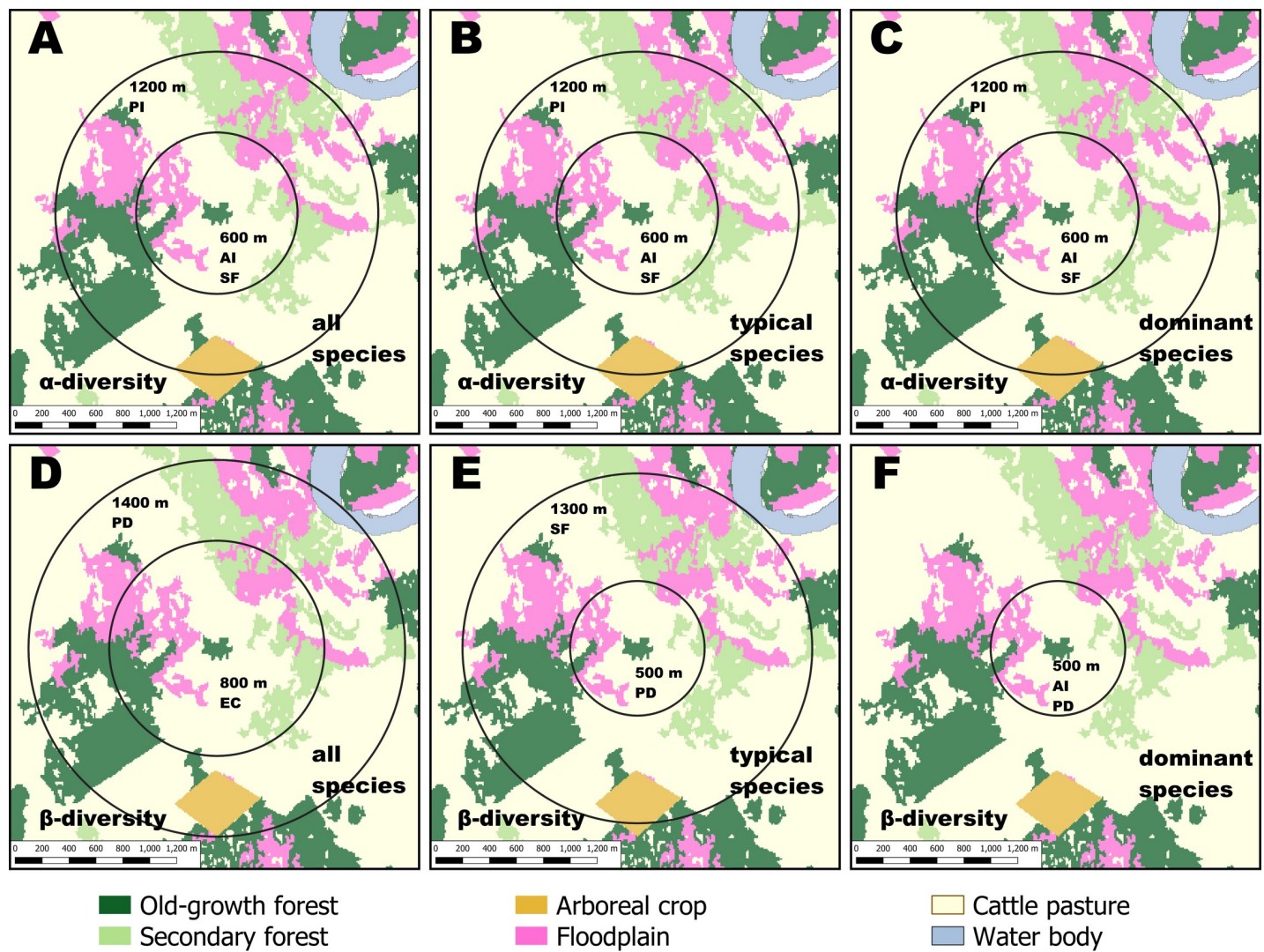

**Fig 2. Scale of effect of landscape metrics on seedling diversity within forest patches in a fragmented tropical rainforest in southern Mexico.** (A-C) Scale of effect on α-diversity. (D-F) Scale of effect on β-diversity. The landscape radius in meters and the associated landscape metrics are within each buffer. The landscape metrics are the aggregation index (AI), the edge contrast index (EC), the patch density (PD), patch isolation (PI), and the percentage of secondary forest (SF).

diversity was the same for typical (807.52 ± 140.03 m; mean ± standard error) and dominant species (801.73 ± 132.57 m), whereas the scale of effect for all species was slightly lower (764.27 ± 144.06 m). The best-fitting models showed a positive effect of patch aggregation, and a negative effect of patch isolation and secondary forest cover (Table I in S1 Text). When assessing the importance and relative effect of these landscape metrics, we found that secondary forest was strongly associated ($\Sigma w_i > 0.75$; Fig 3A) to α-diversity loss, regardless diversity metrics (Fig 3C). We also found an important contribution of patch aggregation in species richness (Fig 3C).

In contrast, we found that scale of effect varied among β-diversity metrics (Fig 2D–2F). The bootstrapped data indicated that the scale of effect decreased in the predicted order (i.e. dominant < typical < all), where the scale of effect was higher for β-diversity of all species (1021.07 ± 133.74 m) than for typical (826.54 ± 84.72 m) and dominant species (577.91 ±101.79 m). The three β-diversity metrics were positively associated with patch

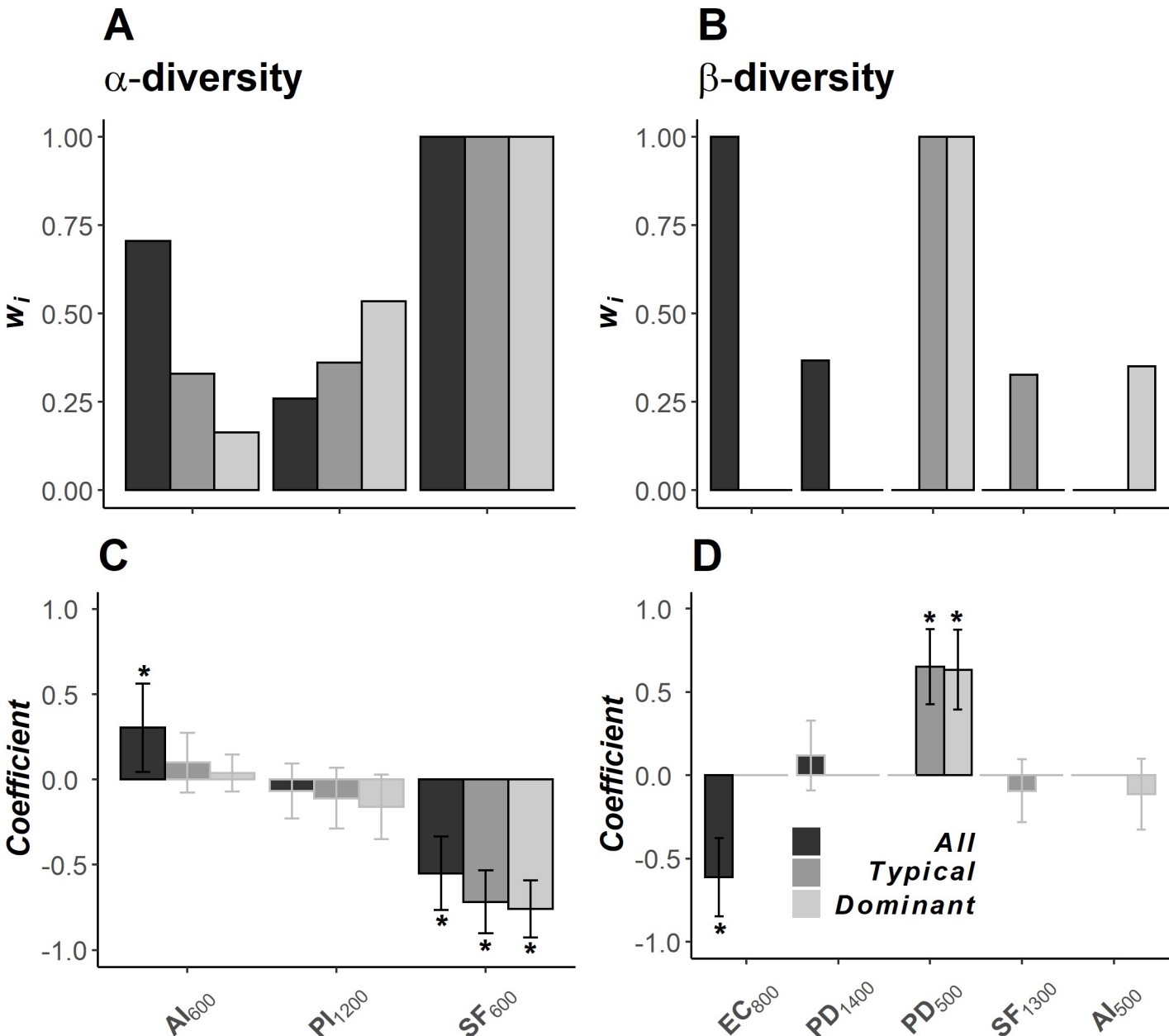

**Fig 3. Importance and relative effects of the landscape metrics on seedling diversity within forest patches in a fragmented tropical rainforest in southern Mexico.** We included the 95% confidence set of models for (A and C) α- and (B and D) β-diversity. The importance of each variable is represented by the sum of the Akaike weights ($\Sigma w_i$). The effects of each covariate were estimated through a model-averaged parameter estimate of information-theoretic-based model selection and multimodel inference. The whiskers represent the unconditional standard error (USE) and the asterisks indicate the influential variables (those for which the USE did not include zero). The landscape metrics are the aggregation index (AI), the edge contrast index (EC), the patch density (PD), patch isolation (PI), and the percentage of secondary forest (SF). The subscript numbers indicate the scale of effect of each variable.

density, and negatively affected by secondary forest cover, patch aggregation, and edge contrast (Table I in S1 Text). The Akaike weighted values showed that landscape configuration was important for β-diversity (Fig 3B). Patch density was particularly relevant for β-diversity of typical and dominant species, while edge contrast was more important for β-diversity of all species (Fig 3D).

## Discussion

### Landscape effects on seedling diversity

This is the first study to assess the effects of landscape structure on the α- and β-diversity of the animal-dispersed tree seedling community at multiple spatial scales (but see [30]). As expected, we found that landscape composition affected α-diversity whereas β-diversity was influenced by landscape configuration. We also observed that the scale of effect varied among β-diversity metrics in the predicted order.

**Effects on α-diversity.**  We found a decrease of α-diversity in forest patches surrounded by secondary forests, which are documented to exert a significant influence in the composition of tree communities within forest patches [32,33,96]. Altered microclimate and the opening of canopy gaps in small patches promote the proliferation of early-successional tree species while reducing the richness and abundance of old-growth species [74,97]. The arrival of early-successional species is driven by short-distance dispersal of generalist seed dispersers [98–100]. This may explain the high abundance of *I. punctata* seedlings, a species from seasonally flooded rainforests that is commonly found in secondary forests and as isolated trees in pastures [101–103]. Additionally, secondary forests restrict the arrival of some forest-specialist seed dispersers to forest patches at similar landscape sizes (100-ha or 564-m radius) [53,104] than the observed scale of effect of secondary forest on α-diversity (600-m radius). Thus, our findings support the contribution of the surrounding matrix on tree diversity within forest patches [32,33]. This results should be taken cautiously, because secondary forests are fundamental for forest regeneration and restoration in HMTLs [105,106], whether by increasing habitat amount, attracting pollinators and seed dispersers, buffering edge effects, and by providing seed sources that maximize the large-scale regeneration of disturbed abandoned fields [107,108]. Thus, conservation strategies in HMTLs should preferably promote secondary forests than agricultural fields in the surrounding matrix of forest patches [105,109].

Contrary to our predictions, we observed that α-diversity was also affected by landscape configuration, supporting the fragmentation threshold hypothesis [38]. The positive effect of patches aggregation on seedling richness suggests that seed sources and seed dispersal can be sustained where patches are highly aggregated. Previous studies predicted that dispersal distance of species may be limited in highly deforested landscapes, favoring species response to local characteristics, such as patches aggregation across the landscape [110,111]. In our study, the selected landscapes presented a low percentage of forest cover across the 13 buffer sizes (19.01±5.1%; mean ± standard deviation). The aggregation of forest patches facilitates the inter-patch movement of forest-dependent species in landscapes with low forest cover, where the distribution of species home range comprises several patches [77,112]. For example, frugivore birds can adjust their movement behavior according to landscape configuration, flying larger distances and visiting more forest patches in landscapes with decreasing forest cover [113]. These changes in the spatial configuration of forest patches also affect species foraging–which relies in resources from nearby patches (i.e. landscape supplementation dynamics [7])–and therefore, seed dispersal patterns. Some studies indicate that fruit removal increased and dispersal distance decreased where plants aggregation is high [114]. Thus, colonization of tree species in highly disturbed landscapes may be operating at smaller spatial scales [42].

**Effects on β-diversity.**  Landscape configuration affected β-diversity, but seedling responses varied among diversity metrics. The β-diversity of all species was negatively associated with edge contrast of old-growth forests in the 800-m radius landscapes. The loss of β-diversity by edge contrast indicates an homogenization of seedling assemblages by edge effects, as well as by a limited heterogeneity of microclimate conditions, a processes observed in HMTLs with low forest cover [18,115]. Additionally, edge contrast can influence the presence

of herbivorous mammals within patches [6], which promote β-diversity of tree seedlings through secondary seed dispersal, and seedling herbivory and trampling [22,23]. Accordingly, the association between seedling β-diversity and edge contrast suggests that low exposure to edge effects, a more heterogeneous microclimate and the presence of density-independent factors (i.e., terrestrial herbivores) promotes a broader species coexistence [35].

The overall increment of β-diversity for the typical and dominant species by patch density is not surprising. According to the "biotic differentiation hypothesis", the limited exchange of seeds and differences in disturbance regimes across forest patches promote floristic differentiation in highly deforested landscapes [68,96]. Spatial patterns that limit the species exchange influence β-diversity in the tropics [17,116]. These effects are intensified in fragmented rainforests, where differences in forest cover amount, disturbance source (i.e., slash-and-burn practices or tree removal by machinery), degree of fragmentation and connectivity, lead to contrasting regeneration dynamics within patches [96,117]. Consequently, the establishment of a wider array of species is facilitated by landscape configuration and a less evenly distributed number of individuals among dominant species [34,77]. These conditions can prevent competitive exclusion (i.e. excluding strong competitors that are weak dispersers), spread the risk of simultaneous extinctions and increase landscape complementation, favoring β-diversity [8,68]. This is consistent with the mechanisms involved in β-diversity increase through the colonization of different opportunistic species and the extinction of shared species among sites [118]. Thus, fragmentation is having positive effects on seedling diversity by facilitating the arrival and establishment of species otherwise poorly represented in old-growth forests [8].

## Scale of landscape effect on diversity metrics

Our results did not support the predicted order of the scale of effect (dominant < typical < all) for α-diversity metrics, which also showed a substantial uncertainty in the selected scale of effect (Fig C in S1 Text). These variations reinforce the findings that scale of effect does not vary in a predictable order among response variables [43,88]. The inconsistencies observed on α-diversity metrics may be associated to the temporal scale and landscape composition and configuration variables assessed [88]. Temporal scale variations regarding tree phenology and movement/foraging patterns of seed dispersers in fragmented HMTLs [54,119] may be obscuring the expected differences in the scale of effect. A study in the same region found that scale of effect of landscape composition on seed rain followed the predicted order [31,60]. The sampling period however, comprised a whole year in order to control for the temporal variation in the seed rain [60]. Additionally, the scale of effect was possibly influenced by the low forest cover of the study area (<25%), which can reduce the expected scale of effect to local factors, such as patch size [42]. However, comparative studies in HMTLs have found non-significant differences in the scale of effect among regions with contrasting disturbance levels [54,60]. Further studies including comparative assessments among regions are needed to fully understand the importance of regional context in landscape metrics and the scale of effect.

On the contrary, the scale of effect observed among β-diversity metrics did follow the predicted order. The scale of effect of edge contrast is similar to those observed for matrix contrast on stem density of understory vegetation in los Tuxtlas, Mexico (798 m), and for matrix openness in seed abundance and species richness of animal-dispersed seeds and forest generalist saplings (798–977 m) [30,31,47]. These studies suggested that matrix contrast operates at smaller scales, where it drives local edge effects involved in the mortality of animal-dispersed seedling species, and favors the dispersal and establishment of wind-dispersed seeds [31,47]. This also holds true for the scale of effect of patch density in our study (500 m) and those observed for richness of old-growth forest specialist- saplings (500 m) in the study area [30].

Patch density is positively related to habitat heterogeneity and to the number of subpopulations in the landscape, increasing community dissimilarity [8,35]. These findings suggest that matrix contrast is limiting the dispersal and recruitment of lower-density species, whereas forest fragmentation is promoting the coexistence of higher-density species.

### Study limitations

Our results are conservative as a larger number of landscapes are desirable to making further inferences. Also, the lack of information related to species composition of secondary forest trees and seed rain within forest patches limit our understanding about the contribution of secondary forests to seedlings α-diversity [60,120]. Finally, our study did not consider a long-term monitoring of tree seedlings to elucidate the contribution of patch microclimate, fruiting events, foraging behavior of seed dispersers, disturbance regimes, and density-dependent mortality factors, which strongly influence the composition and dynamics of seedling communities in the tropics [16,121,122].

For seedlings β-diversity, our findings should be taken cautiously, because we inferred the presence of mammals only through landscape metrics, regardless mammal species have different responses to landscape structure [6]. Furthermore, herbivorous mammal effects on seedling diversity vary according to the incidence, relative abundance and body mass [23,24], as well as by population fluctuations and food quality and availability [123]. In turn, these biological responses (i.e. abundance, richness, incidence and body size) are affected by landscape structure at varying spatial scales [39,42]. Finally, the mechanisms of β-diversity loss are highly sensitive to the ratio of winner-loser species and the spatial scale at which communities are assessed [118]. Nevertheless, the use of structural equation modeling in multi-scale assessments have permitted analyzing the direct and cascading effects of landscape structure on different biological responses in HMTLs [47]. This represents a promising approach to evaluate the response of interactions-mediated patterns and processes to landscape composition and configuration. Thus, further studies regarding cascading effects of landscape structure on different biological responses of herbivorous mammals and tree seedling communities are needed to fill these knowledge gaps.

## Conclusions and conservation implications

Contrary to studies performed in the same area [30], we found that forest fragmentation is a strong driver of seedling diversity at smaller spatial scales in highly deforested rainforests. Severe fragmentation of rainforests will occur in the short term [4], urging economic incentives and innovations in policy and governance that favor forest regrowth, matrix quality and patches number.

Implementation of payment for ecosystem services' schemes and REDD+ programs can prevent further loss of old-growth and secondary forests while promoting agroforestry management [124,125]. Favoring matrix quality and configurational patterns that increase habitat heterogeneity are feasible in landscapes with land-sharing schemes of smallholder agriculture [126]. In these rural landscapes, diversification of small-scale agroecosystems, intensification of agriculture and reduction of cattle ranching promotes forest regeneration and food security [127,128]. Additionally, enactment of programs of enrichment planting of trees with local commercial/ecological importance, and forest management policies that permit local governance for harvesting timber and non-timber products are needed to enhance the management and diversity of secondary forests [129]. Finally, is necessary to map and classify the biophysical and socio-ecological land use dimensions of land covers and naturally regenerated forests to identify restoration opportunities for assisted or unassisted natural regeneration [129,130].

Novel institutional and policy approaches in Costa Rica have proven that decreasing meat exports while supporting payment for environmental services, agricultural intensification and native species plantations lead to a forest transition in less than 15 years [131,132]. Incorporation of these conservation, management and restoration guidelines are urgently needed to preserve regeneration of forest patches and future conservation value of HMTLs.

## Supporting information

**S1 Text. Supporting file.**
(DOCX)

## Acknowledgments

We thank to the people of Quiringüicharo for their hospitality and support. We acknowledge the logistical and technical support provided by J. Manuel Lobato-García during the fieldwork. We thank Gilberto Jamangapé and Rafael Lombera for plant identification. Natura y Ecosistemas Mexicanos A.C. and Arca de Noe provided logistical assistance and accommodation. Adriana L. Luna-Nieves, Bianca A. Santini and four anonymous reviewers provided very valuable comments on earlier versions of the manuscript. SNA is a doctoral student from the Programa de Doctorado en Ciencias Biomédicas, Universidad Nacional Autónoma de México (UNAM), and Consejo Nacional de Ciencia y Tecnología (CONACyT).

## Author Contributions

**Conceptualization:** Sergio Nicasio-Arzeta, Julieta Benítez-Malvido.

**Data curation:** Sergio Nicasio-Arzeta.

**Formal analysis:** Sergio Nicasio-Arzeta, Isela E. Zermeño-Hernández.

**Funding acquisition:** Julieta Benítez-Malvido.

**Investigation:** Sergio Nicasio-Arzeta, Julieta Benítez-Malvido.

**Methodology:** Sergio Nicasio-Arzeta.

**Project administration:** Julieta Benítez-Malvido.

**Resources:** Julieta Benítez-Malvido.

**Software:** Sergio Nicasio-Arzeta.

**Supervision:** Julieta Benítez-Malvido.

**Validation:** Sergio Nicasio-Arzeta, Julieta Benítez-Malvido.

**Visualization:** Sergio Nicasio-Arzeta.

**Writing – original draft:** Sergio Nicasio-Arzeta, Susana Maza-Villalobos.

**Writing – review & editing:** Sergio Nicasio-Arzeta, Isela E. Zermeño-Hernández, Susana Maza-Villalobos, Julieta Benítez-Malvido.

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
