## [Decision Letter · Decision Letter 0]

23 Apr 2021

PONE-D-21-06444

Landscape structure shapes the diversity of tree seedlings at multiple spatial scales in a fragmented tropical rainforest

PLOS ONE

Dear Dr. Nicasio-Arzeta,

Thank you for submitting your manuscript to PLOS ONE. After careful consideration, we feel that it has merit but does not fully meet PLOS ONE’s publication criteria as it currently stands. Therefore, we invite you to submit a revised version of the manuscript that addresses the points raised during the review process.

Both reviewers point out study quality but there are still some points to be considered before an acceptance. Congratulations on your work! 

We look forward to receiving your revised manuscript.

Kind regards,

Juliana Hipólito, Phd

Academic Editor

PLOS ONE

Journal Requirements:

3.1.    You may seek permission from the original copyright holder of Figure 1 to publish the content specifically under the CC BY 4.0 license. 

3.2.    If you are unable to obtain permission from the original copyright holder to publish these figures under the CC BY 4.0 license or if the copyright holder’s requirements are incompatible with the CC BY 4.0 license, please either i) remove the figure or ii) supply a replacement figure that complies with the CC BY 4.0 license. Please check copyright information on all replacement figures and update the figure caption with source information. If applicable, please specify in the figure caption text when a figure is similar but not identical to the original image and is therefore for illustrative purposes only.

Reviewers' comments:

Reviewer's Responses to Questions

**Comments to the Author**

1. Is the manuscript technically sound, and do the data support the conclusions?

Reviewer #1: Yes

Reviewer #2: Partly

2. Has the statistical analysis been performed appropriately and rigorously? 

Reviewer #1: Yes

Reviewer #2: Yes

3. Have the authors made all data underlying the findings in their manuscript fully available?

Reviewer #1: Yes

Reviewer #2: Yes

4. Is the manuscript presented in an intelligible fashion and written in standard English?

Reviewer #1: Yes

Reviewer #2: Yes

5. Review Comments to the Author

Reviewer #1: Review PONE-D21-06444

The “Landscape structure shapes the diversity of tree seedlings at multiple spatial scales in

the fragmented tropical rainforest” brings important data and discussions on the effects of replacing forests with other lower quality environments for plants and seed dispersing animals on recruiting plants in the patches of old-growth forests in tropical forest landscapes. In addition, the study aims to present which spatial scales are most influential in the effects of these replacements and for environmental protection and regeneration actions. The results of this study are very interesting and the discussion is well based on the literature. The introduction clearly sets the objective. The methods and results are well presented. And the whole text is well constructed. Here are some comments that may help improve the study.

Major suggestions:

1.388-391. The suggestion that the secondary forest has a negative effect on the recruitment of mature forests can be misunderstood. Since many reforestation and forest regeneration initiatives attest to the secondary forest as an aid in attracting disperses and pollinators, bridging more impoverished landscapes to landscapes closer to a forest ecosystem. In the sequence, in l.391-392, you use the argument that the composition of the matrix influences recruitment in mature forests. I suggest greater care to indicate the negative effects of the matrix of modified environments on recruitment, and the positive and also the negative effects of the secondary forest. I agree with the quality gradient that you present in l.228-232, in which the secondary forest appears as a link between lower quality environments and the mature forest. I suggest you must include this perspective in the discussion of this argument.

1.423. The explanation of how the different landscape metrics measured in the study may reflect the presence of seed dispersing animals is clear, but this is an indirect inference. As you have not measured the presence of mammals in these landscapes, it may be interesting to bring to the end of this paragraph the aspects observed in this study, such as the effects of different types of environment on beta diversity. Thus, the explanation of the effects of the contrast between the edges of different environments can be made clearer.

1.509-511. Although the observed is different from the expected, you still have an interesting and important argument to promote conservation actions and future studies. The scenario of fragmentation progress and at an accelerated pace in tropical landscapes is frightening and information such as that generated in studies like this should be valued and better explored. I suggest that this concluding section and recommendations give more prominence to the findings of this study and to the recommendations that can be transformed into protective landscape policies and that can promote the recovery of native forests, considering all the advantages of the native forest for human sustainable development.

1.500. The conclusion is long and a “take home message” more effective is needed to related the findings of this study.

Minor suggestions:

l.32-33. I suggest another presentation of the argument about the secondary forest, see major suggestions.

l.36. Remove the zero before “0findings”.

l.38-39. The end of the abstract is very vague. I suggest ending this section with the “take home message” from this study. What is the main message of the study? Include that message here.

l.49. I suggest better specifying what types of land use changes you are referring to here, to increase the clarity of the argument.

1.56-57. I suggest rewriting this sentence, the idea of increasing and maintaining biodiversity or species diversity is repeated.

l.60-61. “And compositional similarity (β-diversity) of the seedling community”, why? I suggest including an explanation of how the beta diversity of the tree community depends on composition at the seedling level. I also suggest specifying whether the alpha diversity is of tree species at the adult or seedling level.

l.61-66. I suggest dividing this sentence, the explanations in parentheses can make reading difficult. I suggest including an explanation for “propagule availability”, “suitable environmental conditions” and “density dependence effects” as the explanations were presented for “vertebrate seed dispersal” and “seedling establishment”.

1.68-69. Do any of these strategies specifically refer to the seedling community? If so, I suggest including a highlight for these publications here. If not, it is one more argument to strengthen this study.

l.70-93. This paragraph is too long. I suggest grouping the effects of habitat loss and fragmentation (here separated in composition and landscape configuration) on both alpha and beta diversity. For example, start with the effects of composition on both components of diversity. And in another paragraph talk about the effects of the landscape configuration in the two components of the diversity of the seedling community. The indirect effects of the composition and configuration of the landscape in the seedling community through vertebrate disperses is brought up several times in the paragraph. I suggest bringing this information only once.

l.94-95. It is not clear here what is meant by “landscape structure”. I suggest clarifying.

l.101-103. It is not clear here why this is expected. It is important to make clear.

l.103-110. This should go to the methods section.

l.133. The symbol of degree Celsius should be close to the number.

l.137. I suggest removing the "however".

l.138. It could include here how long this forest loss occurred to justify “rapid deforestation”.

l.139-140. You should include here a description of how the forest patches were selected and why.

l.188. Removes an extra point in the sentence.

l.227. Why only for mammals and not for mammals and birds. Please explain.

1.243. Replace the dash with a space in the number “4000-m”.

1.264. What does “split of the data set” mean, please explain.

l.276. Add a space before the parentheses.

l.281. Replace “re-sampled” with resampled.

l.321. Include a space after the dash.

l.322. Remove the space after the number “33”.

l.336. Use the name Secondary forest in full here to be consistent with the names of the variables you are using in this paragraph.

l.349. I suggest including a figure with emphasis on the scales of effect for each variable, as well as in figure one, but only with the buffer of the indicated the effect distance.

l.354. This table is difficult to read. I suggest including a first column with the variables that appear in each model (as in model syntax in R) followed by the columns that show R2, AICc, ∆ and wi. It is not clear the values presented for each explanatory variable.

1.358-363. This paragraph should come in the title of the caption.

1.362-363. I suggest including another table to present these values.

1.406. Add a space between the parentheses and the dash.

1.526-527. Please include secondary forest in this list.

l.918. In the figure it is missing to include the reference black color in the legend inside the figure.

Reviewer #2: GENERAL COMMENTS

This article is very interesting and well done. It is impressive the amount of work behind this research, and I want to congratulate the authors. The introduction is appropriate and nicely presents the manuscript. The methods and statistics are solid and the ones used in the more recent landscape researches. The results are clear and relevant, although some reordering is suggested. The discussion section is pertinent, however, there are some ecological filters that the authors should discuss. In this sense, I give some recommendations to the authors hoping to improve this already good manuscript.

How did the authors know the species they found were dispersed or from the trees above the plots? The Discussion is heavily focused on dispersal limitation, however, they are working with seedlings, not seeds. This stage involves not just seed arrival to the site but a successful establishment in the site. Large-distance seed dispersal is not very often, the most common events are short-distance dispersal. This means that most species come from nearby trees, particularly the small ones that could be the result of the last seed production for that species (e.g. Inga). Inga spp have recalcitrant seeds with no dormancy, which means the small seedlings you measured were probably from the last seed production. Also, the small ones might be more related to seed dispersal processes but also for seed production the last year. The larger ones are the result of the survival to establishment limitations and may reflect a longer temporal process of seed dispersal and differential establishment.

The author also did not mention that tropical rainforests are highly beta-diverse in space, with most species being very rare. This characteristic of the rainforests makes them highly species-rich.

MINOR COMMENTS

ABSTRACT

Lines 25-26. But see Arasa-Gisbert et al. 2021 J Ecol. Here the authors should address this study that just came out and it is made with sapling assemblages (≥30 cm height and <1 cm diameter) in old‐growth forest fragments also in the Lacandona region. This way, comparing the results with this study would greatly improve the discussion section.

Line 36. Our 0findings

Lines 36-38. The authors did not measure seed dispersal but seedling assemblages, so other processes such as establishment limitations should be addressed.

Lines 38-39. The conclusion is somehow something we already knew from previous studies and too general, it should be more focused on the particular findings of the study.

INTRODUCTION

Lines 62-63. It is not correct to consider saplings and seedlings as propagules. Authors should review the propagule definition

Line 65. Remove “vertebrates”

Line 67. Disruption of these processes by fragmentation?

Line 72. Add often, as you mention studies at the landscape scale “…research on tree seedlings is often limited…”

Lines 75-77. Landscape composition has strong effects also on seed sources, not just seed dispersal.

Line 82. There is a typo “given by of forest”

Lines 83-85. This article talks about the effects of deforestation and isolation, both associated with habitat loss, not fragmentation. Also, see Arasa-Gisbert et al. 2021 who did not find the effects of fragmentation.

Lines 101-103. This sentence is unclear to me, maybe some rephrasing?

Line 116. Remove “however”

Line 118. But see Arasa-Gispert et al. 2021

Line 123. The authors should state clearly the hypothesis and predictions (i.e. expected results). The predictions as written, do not have a direction so they are ambiguous. Diversity is going to be influenced by the landscape, but how? magnitude? Also, the authors should mention which landscape metrics are using to measure landscape composition and landscape configuration. In Table 1. Ecological relevance, the authors give some expected results for the metrics they are testing that could be useful for the hypothesis/predictions section. As a suggestion, authors could use Table 1 to state the predictions associated with each variable.

METHODS

Line 134. “precipitation” instead of “precipitations”

Line 164. Remove “and” or “then”

Line 178. of order…? It seems something is lacking here

Line 188. There are two dots. Use superscript in m2

Line 222. the mean distance among all forest patches in the landscape (patch isolation; PI),

Table 1. Secondary forest is also an indicator of the landscape-scale habitat amount. It is positively associated with the availability of propagules and seed dispersal. Also, it is not completely clear how the authors differentiate between SF and FC, until what age a forest was considered second-growth?

Table 1. Patch density has not been associated with floristic differentiation. The study from Arroyo-Rodríguez et al. found differentiation was related to a highly deforested landscape with high isolation among patches.

Line 244. It seems there is a typo in “(of a 300…”

Lines 246-253. This part is a little bit confusing to me, maybe some rewording is needed. The authors first mention there is some autocorrelation among closer sites, but then state there is no spatial autocorrelation.

Table 2. The caption should indicate what AI, PI, EC, SV, PD stand for. Also the subindices. SV = SF?

RESULTS

Line 322. There is a space “33 %”.

Tables are in a different format

Lines 358-363. This should go in Table 4 caption

Table 4. This could go to supplementary

DISCUSSION

Line 369. But see Arasa-Gisbert et al. 2021

Lines 377-379. This explanation is not very informative. What kind of influence? What are the long-time scales?

Line 391. Please give a reference for this statement: which are favored by the environmental conditions (e.g., increased light and temperature) within forest patches.

Line 399. Aggregation was not calculated at patch but at the landscape scale.

Lines 415-417. The authors did not assessed dispersal limitation (see Muller–Landau et al. 2002- Assessing Recruitment Limitation: Concepts, Methods and Case-studies from a Tropical Forest).

Lines 395-397. And seed sources and successful establishment

Line 398. What does a highly disturbed landscape refer to?

Line 429 What does disturbance type refer to?

Lines 430-432. The phrase is not accurate. Which are the species favored by the patch density (fragmentation)? Life-history traits are directly associated with a resource strategy, evolution phylogeny, which are dependent on the environment. Then, there are species favored by changes in abiotic conditions (e.g. light) on edges, canopy gaps, etc.

Line 433. What are the landscape conditions? Are landscape conditions = landscape structure?

Line 450. Landscape context? Please use the same terms across the manuscript.

Line 521. Secondary forests could be highly heterogeneous (see Chazdon et al. publications)

REFERENCES

Line 586. last names in capital

Line 624. Hölzel N, editor.

Line 663. Kitzberger T, editor

Line 664. There are squares among the last names, must be some symbol issue

Line 700. Format: capitals in the title

Line 791. Format: capitals in the title

Line 871. Format: capitals in the title

FIGURES

Line 922. Please add here a brief explanation for beta diversity as for the other ones. It was measured among plots or fragments? “B-diversity within a forest fragment"

6. PLOS authors have the option to publish the peer review history of their article (what does this mean?). If published, this will include your full peer review and any attached files.

Reviewer #1: No

Reviewer #2: No

---

## [Author Response · Author response to Decision Letter 0]

17 May 2021

EDITOR. Thank you for submitting your manuscript to PLOS ONE. After careful consideration, we feel that it has merit but does not fully meet PLOS ONE’s publication criteria as it currently stands. Therefore, we invite you to submit a revised version of the manuscript that addresses the points raised during the review process.

Both reviewers point out study quality but there are still some points to be considered before an acceptance. Congratulations on your work!

Authors. We are grateful for the careful review performed by the reviewers and yourself

JOURNAL REQUIREMENTS

Authors: Corrected as suggested

Authors: We will provide the repository information and DOI in the data availability statement

Authors: The copyright belongs to Sergio Nicasio-Arzeta, the first and corresponding author of this work. This copyright belongs from a preprint uploaded in October 31st, 2019 (https://doi.org/10.1101/826339) by myself. Therefore, I grant permission to myself to publish Figure 1 in my study. Additionally, I confirm that the original map image used to create both Figure 1 and Figure 2 was created by the corresponding author of this manuscript, Sergio Nicasio-Arzeta. You can verify the name and ORCID of the corresponding author of the preprint manuscript, which is Sergio Nicasio-Arzeta: https://www.biorxiv.org/content/10.1101/826339v1. Finally, I declare that corresponding author of this manuscript and the preprint version where the original map image is stored is the same person, whose name is Sergio Nicasio-Arzeta. The adapted published figure is attached as "Original figure.tif". The image appears in Fig 1, Materials and methods section, "Study site" subsection, page six. Also, the image appears in Fig 2, Results section, "Scale of effect and importance of landscape structure on seedling diversity" subsection, page 17.

Authors: Corrected as suggested

REVIEWER COMMENTS

GENERAL COMMENTS--------------------------------------

REVIEWER 1. The “Landscape structure shapes the diversity of tree seedlings at multiple spatial scales in the fragmented tropical rainforest” brings important data and discussions on the effects of replacing forests with other lower quality environments for plants and seed dispersing animals on recruiting plants in the patches of old-growth forests in tropical forest landscapes. In addition, the study aims to present which spatial scales are most influential in the effects of these replacements and for environmental protection and regeneration actions. The results of this study are very interesting and the discussion is well based on the literature. The introduction clearly sets the objective. The methods and results are well presented. And the whole text is well constructed. Here are some comments that may help improve the study.

REVIEWER 2. This article is very interesting and well done. It is impressive the amount of work behind this research, and I want to congratulate the authors. The introduction is appropriate and nicely presents the manuscript. The methods and statistics are solid and the ones used in the more recent landscape researches. The results are clear and relevant, although some reordering is suggested.

Authors: Thank you very much for this positive feedback

ABSTRACT--------------------------------------------------------

REVIEWER 2

Lines 25-26. But see Arasa-Gisbert et al. 2021 J Ecol. Here the authors should address this study that just came out and it is made with sapling assemblages (≥30 cm height and <1 cm diameter) in old‐growth forest fragments also in the Lacandona region. This way, comparing the results with this study would greatly improve the discussion section.

Authors: Corrected as suggested (line 27)

REVIEWER 1

Lines 32-33. I suggest another presentation of the argument about the secondary forest, see major suggestions.

Authors: Corrected as suggested (line 33)

Line 36. Remove the zero before “0findings”.

Authors: Corrected as suggested

REVIEWER 2

Line 36. Our 0findings

Authors: Corrected as suggested

Lines 36-38. The authors did not measure seed dispersal but seedling assemblages, so other processes such as establishment limitations should be addressed.

Authors: We rephrased it by including a “take home message” (lines 35-36)

REVIEWER 1

Lines 38-39. The end of the abstract is very vague. I suggest ending this section with the “take home message” from this study. What is the main message of the study? Include that message here.

REVIEWER 2

Lines 38-39. The conclusion is somehow something we already knew from previous studies and too general, it should be more focused on the particular findings of the study.

Authors: We rephrased it by including the main message of the study, which is the role of landscape configuration on sustaining seedling diversity (lines 36-40)

INTRODUCTION----------------------------------------------------

REVIEWER 1

Line 49. I suggest better specifying what types of land use changes you are referring to here, to increase the clarity of the argument.

Authors: Corrected as suggested as “forest conversion to cattle pastures and large-scale plantations” (line 49)

Lines 56-57. I suggest rewriting this sentence, the idea of increasing and maintaining biodiversity or species diversity is repeated.

Authors: Corrected as “Thus, sustaining forest regeneration within patches is critical to promote the long-term maintenance of biodiversity in HMTLs” (lines 57-58)

Lines 60-61. “And compositional similarity (β-diversity) of the seedling community”, why? I suggest including an explanation of how the beta diversity of the tree community depends on composition at the seedling level. I also suggest specifying whether the alpha diversity is of tree species at the adult or seedling level.

Authors: We specified that alpha diversity is of seedling level (line 61). Also, we briefly explained what happens to the tree community when alpha and beta diversity of seedlings decreases (lines 63-68)

REVIEWER 2

Lines 62-63. It is not correct to consider saplings and seedlings as propagules. Authors should review the propagule definition.

Authors: We deleted it to avoid confusions

Line 65. Remove “vertebrates”

Authors: Removed as suggested

Lines 61-66. I suggest dividing this sentence, the explanations in parentheses can make reading difficult. I suggest including an explanation for “propagule availability”, “suitable environmental conditions” and “density dependence effects” as the explanations were presented for “vertebrate seed dispersal” and “seedling establishment”.

Authors: We rewrote this sentence (lines 61-66)

REVIEWER 2

Line 67. Disruption of these processes by fragmentation?

Authors: Overall disruption (that includes defaunation). We rephrased the line to avoid linking the lack of seed dispersal, environmental heterogeneity and density-dependence effects to a particular type of disturbance (i.e., defaunation, forest loss/fragmentation), because we only wanted to mention the consequences of seedling alpha and beta diversity loss (lines 63-68)

REVIEWER 1

Lines 68-69. Do any of these strategies specifically refer to the seedling community? If so, I suggest including a highlight for these publications here. If not, it is one more argument to strengthen this study.

Authors: We rephrased the lines to mention the lack of strategies for seedling communities in HMTLs (lines 68-72)

Lines 70-93. This paragraph is too long. I suggest grouping the effects of habitat loss and fragmentation (here separated in composition and landscape configuration) on both alpha and beta diversity. For example, start with the effects of composition on both components of diversity. And in another paragraph talk about the effects of the landscape configuration in the two components of the diversity of the seedling community. The indirect effects of the composition and configuration of the landscape in the seedling community through vertebrate disperses is brought up several times in the paragraph. I suggest bringing this information only once.

Authors: Corrected by grouping the effects of landscape structure on alpha (lines 74-79) and beta diversity (lines 79-88)

REVIEWER 2

Line 72. Add often, as you mention studies at the landscape scale “…research on tree seedlings is often limited…”

Authors: We deleted this phrase to avoid departing from the main message of the study (lines 68-72)

Lines 75-77. Landscape composition has strong effects also on seed sources, not just seed dispersal.

Authors: Corrected as suggested (lines 74-78)

Line 82. There is a typo “given by of forest”

Authors: Corrected (lines 82-83)

Lines 83-85. This article talks about the effects of deforestation and isolation, both associated with habitat loss, not fragmentation. Also, see Arasa-Gisbert et al. 2021 who did not find the effects of fragmentation.

Authors: Corrected as suggested (lines 78-79)

REVIEWER 1

Lines 94-95. It is not clear here what is meant by “landscape structure”. I suggest clarifying.

Authors: Clarified as suggested (lines 89-90)

Lines 101-103. It is not clear here why this is expected. It is important to make clear.

Authors: Clarified as suggested (lines 96-98)

REVIEWER 2

Lines 101-103. This sentence is unclear to me, maybe some rephrasing?

Authors: Clarified as suggested (lines 96-98)

REVIEWER 1

Lines 103-110. This should go to the methods section.

Authors: We moved these lines to the methods section (lines 166-170)

REVIEWER 2

Line 116. Remove “however”

Authors: Removed as suggested (line 104)

Line 118. But see Arasa-Gispert et al. 2021

Authors: Corrected as suggested (line 106)

Line 123. The authors should state clearly the hypothesis and predictions (i.e. expected results). The predictions as written, do not have a direction so they are ambiguous. Diversity is going to be influenced by the landscape, but how? magnitude? Also, the authors should mention which landscape metrics are using to measure landscape composition and landscape configuration. In Table 1. Ecological relevance, the authors give some expected results for the metrics they are testing that could be useful for the hypothesis/predictions section. As a suggestion, authors could use Table 1 to state the predictions associated with each variable.

Authors: Corrected as suggested (lines 111-114)

MATERIALS AND METHODS-------------------------------------------

REVIEWER 1

Line 133. The symbol of degree Celsius should be close to the number.

Authors: Corrected as suggested (line 121)

REVIEWER 2

Line 134. “precipitation” instead of “precipitations”

Authors: Corrected as suggested (line 122)

REVIEWER 1

Line 137. I suggest removing the "however".

Authors: Corrected as suggested (line 124)

Line 138. It could include here how long this forest loss occurred to justify “rapid deforestation”.

Authors: Corrected as suggested (line 126)

Lines 139-140. You should include here a description of how the forest patches were selected and why.

Authors: Patches description and the criteria employed to select them are explained in lines 127-131

REVIEWER 2

Line 164. Remove “and” or “then”

Authors: Corrected as suggested (line160)

Line 178. of order…? It seems something is lacking here

Authors: Corrected as suggested (line 179)

REVIEWER 1

Line 188. Removes an extra point in the sentence.

Authors: Corrected as suggested (line192)

REVIEWER 2

Line 188. There are two dots. Use superscript in m2

Authors: Corrected as suggested (line190)

Line 222. the mean distance among all forest patches in the landscape (patch isolation; PI)

Authors: Corrected as suggested (line 224)

Table 1. Secondary forest is also an indicator of the landscape-scale habitat amount. It is positively associated with the availability of propagules and seed dispersal. Also, it is not completely clear how the authors differentiate between SF and FC, until what age a forest was considered second-growth?

Authors: We considered second growth those shrub and tree vegetation up to 15 years of abandonment (the maximum abandonment age during the sampling period), which were differentiated from old-growth forest in the supervised classification. We agree that secondary forests provide of propagules and seeds, however, these forests usually harbor a lesser proportion of animal-dispersed tree species that their old-growth forests counterpart

Table 1. Patch density has not been associated with floristic differentiation. The study from Arroyo-Rodríguez et al. found differentiation was related to a highly deforested landscape with high isolation among patches.

Authors: Corrected (Table 1)

REVIEWER 1

Line 227. Why only for mammals and not for mammals and birds. Please explain.

Authors: Because bird persistence within forest patches is more related with forest cover amount than matrix composition (Carrara E, Arroyo-Rodríguez V, Vega-Rivera JH, Schondube JE, de Freitas SM, Fahrig L. Impact of landscape composition and configuration on forest specialist and generalist bird species in the fragmented Lacandona rainforest, Mexico. Biol Conserv. 2015;184: 117–126). Additionally, since forest mammals are sensitive to the amount of tree vegetation cover in the surrounding landscape, the quality values employed are also useful for arboreal primates (Galán-Acedo C, Arroyo-Rodríguez V, Estrada A, Ramos-Fernández G. Forest cover and matrix functionality drive the abundance and reproductive success of an endangered primate in two fragmented rainforests. Landsc Ecol. 2019;34: 147–158), which are important seed dispersers in the study area. 

Line 243. Replace the dash with a space in the number “4000-m”.

Authors: Corrected as suggested (line 246)

REVIEWER 2

Line 244. It seems there is a typo in “(of a 300…”

Authors: Corrected as suggested (line 247)

Lines 246-253. This part is a little bit confusing to me, maybe some rewording is needed. The authors first mention there is some autocorrelation among closer sites, but then state there is no spatial autocorrelation.

Authors: We mentioned correlation between two landscape metrics (AI and FC) at larger buffer sizes (lines 248-249). We rephrased this paragraph to make clear that no spatial autocorrelation was found between sites and diversity metrics or between sites and landscape metrics (lines 248-254).

Table 2. The caption should indicate what AI, PI, EC, SV, PD stand for. Also, the subindices. SV = SF?

Authors: Corrected as suggested (Table 2)

REVIEWER 1

Line 264. What does “split of the data set” mean, please explain.

Authors: Explained as suggested (262-266)

Line 276. Add a space before the parentheses.

Authors: Corrected as suggested (line 278)

Line 281. Replace “re-sampled” with resampled.

Authors: Corrected as suggested (line 283)

RESULTS----------------------------------------------------------

REVIEWER 1

Lines 321. Include a space after the dash.

Authors: Corrected as suggested (line 327)

Lines 322. Remove the space after the number “33”.

Authors: Corrected as suggested (line 329)

REVIEWER 2

Line 322. There is a space “33 %”.

Authors: Corrected as suggested (line 329)

Tables are in a different format

Authors: Corrected as suggested

REVIEWER 1

Lines 336. Use the name Secondary forest in full here to be consistent with the names of the variables you are using in this paragraph.

Authors: Corrected as suggested (line 343)

Lines 349. I suggest including a figure with emphasis on the scales of effect for each variable, as well as in figure one, but only with the buffer of the indicated the effect distance.

Authors: Included as suggested (Figure 2)

Lines 354. This table is difficult to read. I suggest including a first column with the variables that appear in each model (as in model syntax in R) followed by the columns that show R2, AICc, ∆ and w It is not clear the values presented for each explanatory variable.

Authors: Corrected as suggested. We also moved this table to the supplementary section (Table I in S1 Text), as suggested by reviewer 2

Lines 358-363. This paragraph should come in the title of the caption.

Authors: Corrected as suggested (Table I in S1 Text)

Lines 362-363. I suggest including another table to present these values.

Authors: We found that magnitude and direction of landscape metrics are well represented in Figure 3. Additionally, we included this information in Table I in S1 Text

REVIEWER 2

Lines 358-363. This should go in Table 4 caption

Authors: Corrected as suggested (Table I in S1 Text)

Table 4. This could go to supplementary

Authors: Corrected as suggested. Thank you, I always thought the same, regardless the insistence for including it in the main text by the previous reviewers. We also corrected the table as suggested by reviewer 1

DISCUSION--------------------------------------------------------

REVIEWER 2

The discussion section is pertinent, however, there are some ecological filters that the authors should discuss. In this sense, I give some recommendations to the authors hoping to improve this already good manuscript.

How did the authors know the species they found were dispersed or from the trees above the plots? The Discussion is heavily focused on dispersal limitation; however, they are working with seedlings, not seeds. This stage involves not just seed arrival to the site but a successful establishment in the site.

Authors: We corrected the discussion to focus on how habitat availability/heterogeneity influence seedling establishment and beta diversity

Large-distance seed dispersal is not very often, the most common events are short-distance dispersal. This means that most species come from nearby trees, particularly the small ones that could be the result of the last seed production for that species (e.g. Inga). Inga spp have recalcitrant seeds with no dormancy, which means the small seedlings you measured were probably from the last seed production.

Authors: We specified that short-distance dispersal influences species arrival from the surrounding matrix (lines 390-393)

Also, the small ones might be more related to seed dispersal processes but also for seed production the last year. The larger ones are the result of the survival to establishment limitations and may reflect a longer temporal process of seed dispersal and differential establishment.

Authors: Contrary to saplings (the larger ones), seedling establishment is strongly influenced to edge effects, microclimate (i.e., light, temperature and humidity) variations and herbivory within patches.

The author also did not mention that tropical rainforests are highly beta-diverse in space, with most species being very rare. This characteristic of the rainforests makes them highly species-rich.

Authors: We mentioned that species coexistence at very low densities makes rainforests highly diverse (lines 98-100)

REVIEWER 2

Line 369. But see Arasa-Gisbert et al. 2021

Authors: Corrected as suggested, regardless Arasa-Gisbert assessed saplings and did not consider β-diversity (line 379)

Lines 377-379. This explanation is not very informative. What kind of influence? What are the long-time scales?

Authors: Corrected to avoid confusions (lines 386-388)

REVIEWER 1

Lines 388-391. The suggestion that the secondary forest has a negative effect on the recruitment of mature forests can be misunderstood. Since many reforestation and forest regeneration initiatives attest to the secondary forest as an aid in attracting disperses and pollinators, bridging more impoverished landscapes to landscapes closer to a forest ecosystem. In the sequence, in l.391-392, you use the argument that the composition of the matrix influences recruitment in mature forests. I suggest greater care to indicate the negative effects of the matrix of modified environments on recruitment, and the positive and also the negative effects of the secondary forest. I agree with the quality gradient that you present in l.228-232, in which the secondary forest appears as a link between lower quality environments and the mature forest. I suggest you must include this perspective in the discussion of this argument.

Authors: We included this perspective in lines 396-402

REVIEWER 2

Line 391. Please give a reference for this statement: which are favored by the environmental conditions (e.g., increased light and temperature) within forest patches.

Authors: We rephrased it to avoid confusions (lines 396-398)

Lines 395-397. And seed sources and successful establishment

Authors: Corrected as suggested (lines 404-406)

Line 398. What does a highly disturbed landscape refer to?

Authors: Corrected as “…highly deforested landscapes…” (line 407)

Line 399. Aggregation was not calculated at patch but at the landscape scale.

Authors: Corrected as “…such as patches aggregation across the landscape” (line 408)

REVIEWER 1

Lines 406. Add a space between the parentheses and the dash.

Authors: Corrected as suggested (line 416)

Lines 415-417. The authors did not assess dispersal limitation (see Muller–Landau et al. 2002- Assessing Recruitment Limitation: Concepts, Methods and Case-studies from a Tropical Forest).

Authors: We rephrased this paragraph to highlight the role of edge effects and landscape heterogeneity on seedling β-diversity (lines 424-426) 

REVIEWER 1

Line 423. The explanation of how the different landscape metrics measured in the study may reflect the presence of seed dispersing animals is clear, but this is an indirect inference. As you have not measured the presence of mammals in these landscapes, it may be interesting to bring to the end of this paragraph the aspects observed in this study, such as the effects of different types of environment on beta diversity. Thus, the explanation of the effects of the contrast between the edges of different environments can be made clearer.

Authors: We included this perspective in lines 424-431

REVIEWER 2

Line 429 What does disturbance type refer to?

Authors: Mentioned in lines 436-438

Lines 430-432. The phrase is not accurate. Which are the species favored by the patch density (fragmentation)? Life-history traits are directly associated with a resource strategy, evolution phylogeny, which are dependent on the environment. Then, there are species favored by changes in abiotic conditions (e.g. light) on edges, canopy gaps, etc.

Authors: We deleted this phrase to avoid further misunderstandings

Line 433. What are the landscape conditions? Are landscape conditions = landscape structure?

Authors: Corrected as “landscape configuration” (line 439)

Line 450. Landscape context? Please use the same terms across the manuscript.

Authors: Corrected as “landscape composition and configuration” (line 454)

REVIEWER 1

Line 500. The conclusion is long and a “take home message” more effective is needed to related the findings of this study.

Authors: We added a “take home message” (lines 504-507)

Lines 509-511. Although the observed is different from the expected, you still have an interesting and important argument to promote conservation actions and future studies. The scenario of fragmentation progress and at an accelerated pace in tropical landscapes is frightening and information such as that generated in studies like this should be valued and better explored. I suggest that this concluding section and recommendations give more prominence to the findings of this study and to the recommendations that can be transformed into protective landscape policies and that can promote the recovery of native forests, considering all the advantages of the native forest for human sustainable development.

Authors: We rewrote the conclusions to focus on policy and scheme recommendations associated to the main findings of our study (lines 508-523)

REVIEWER 2

Line 521. Secondary forests could be highly heterogeneous (see Chazdon et al. publications)

Authors: We deleted this phrase to avoid confusions

REVIEWER 1

Lines 526-527. Please include secondary forest in this list.

Authors: Included as suggested (line 509)

REFERENCES-------------------------------------------------------

REVIEWER 2

Line 586. last names in capital

Authors: Corrected as suggested (line 559)

Line 624. Hölzel N, editor.

Authors: Corrected as suggested (line 851)

Line 663. Kitzberger T, editor

Authors: Corrected as suggested (line 710)

Line 664. There are squares among the last names, must be some symbol issue

Authors: That does not appear in my file, must be some symbol configuration

Line 700. Format: capitals in the title

Authors: Corrected as suggested (line 654)

Line 791. Format: capitals in the title

Authors: Corrected as suggested (line 744)

Line 871. Format: capitals in the title

Authors: Corrected as suggested (line 842)

FIGURES-------------------------------------------------------

REVIEWER 1

Lines 918. In the figure it is missing to include the reference black color in the legend inside the figure.

Authors: We included that reference in the new figure 1

REVIEWER 2

Line 922. Please add here a brief explanation for beta diversity as for the other ones. It was measured among plots or fragments? “B-diversity within a forest fragment"

Authors: Explained as suggested (line 138)

---

## [Editor Report · Decision Letter 1]

2 Jun 2021

Landscape structure shapes the diversity of tree seedlings at multiple spatial scales in a fragmented tropical rainforest

PONE-D-21-06444R1

Dear Dr. Nicasio-Arzeta,

We’re pleased to inform you that your manuscript has been judged scientifically suitable for publication and will be formally accepted for publication once it meets all outstanding technical requirements.

Kind regards,

Juliana Hipólito, Phd

Academic Editor

PLOS ONE
---

## [Editor Report · Acceptance letter]

24 Jun 2021

PONE-D-21-06444R1 

Landscape structure shapes the diversity of tree seedlings at multiple spatial scales in a fragmented tropical rainforest 

Dear Dr. Nicasio-Arzeta:

I'm pleased to inform you that your manuscript has been deemed suitable for publication in PLOS ONE. Congratulations! Your manuscript is now with our production department. 

Kind regards, 

on behalf of

Dr. Juliana Hipólito 

Academic Editor

PLOS ONE